# A Machine Learning Model for Predicting Sit-to-Stand Trajectories of People with and without Stroke: Towards Adaptive Robotic Assistance

**DOI:** 10.3390/s22134789

**Published:** 2022-06-24

**Authors:** Thomas Bennett, Praveen Kumar, Virginia Ruiz Garate

**Affiliations:** 1Bristol Robotics Laboratory, Faculty of Environment and Technology, University of the West of England, Bristol BS16 1QY, UK; thomas10.bennett@uwe.ac.uk; 2Faculty of Health and Applied Sciences, University of the West of England, Bristol BS16 1DD, UK; praveen.kumar@uwe.ac.uk

**Keywords:** gait analysis, machine learning, stroke, rehabilitation, robotics, wearable sensors, dataset

## Abstract

Sit-to-stand and stand-to-sit transfers are fundamental daily motions that enable all other types of ambulation and gait. However, the ability to perform these motions can be severely impaired by different factors, such as the occurrence of a stroke, limiting the ability to engage in other daily activities. This study presents the recording and analysis of a comprehensive database of full body biomechanics and force data captured during sit-to-stand-to-sit movements in subjects who have and have not experienced stroke. These data were then used in conjunction with simple machine learning algorithms to predict vertical motion trajectories that could be further employed for the control of an assistive robot. A total of 30 people (including 6 with stroke) each performed 20 sit-to-stand-to-sit actions at two different seat heights, from which average trajectories were created. Weighted *k*-nearest neighbours and linear regression models were then used on two different sets of key participant parameters (height and weight, and BMI and age), to produce a predicted trajectory. Resulting trajectories matched the true ones for non-stroke subjects with an average $R^{2}$ score of 0.864±0.134 using *k* = 3 and 100% seat height when using height and weight parameters. Even among a small sample of stroke patients, balance and motion trends were noticed along with a large within-class variation, showing that larger scale trials need to be run to obtain significant results. The full dataset of sit-to-stand-to-sit actions for each user is made publicly available for further research.

## 1. Introduction

Stroke occurs when there is lack of blood supply to the brain and can cause a wide variety of physical, psychological and cognitive problems, leading to long-term disability [1]. According to the World Stroke Organisation, over 13 million people have a stroke each year (https://www.world-stroke.org/, accessed on 26 May 2022). In the UK alone, there are estimated to be around 1.3 million stroke survivors, with instances potentially increasing 60% from 2015 to 2035 [2]. The financial burden of stroke rehabilitation on healthcare services is also rising rapidly, which, combined with the problem of staff shortages (https://nhsfunding.info/symptoms/10-effects-of-underfunding/staff-shortages, accessed on 26 May 2022), increases the need to consider technological solutions for supporting people with stroke.

Robotics has emerged as a promising beacon in physical assistance to address service shortage and support healthcare workers in the care and rehabilitation of people with stroke. Mobility assistive robotic devices include (but are not restricted to) smart walkers [3] and sit-to-stand (STS) aids [4], which help people who need physical assistance to perform activities of daily living (ADLs). ADLs are defined as self-care tasks, such as bathing, dressing, toileting, transferring, and feeding [5]. Transfers such as getting in and out of bed, chair or toilet involve STS.

STS activities are thus key for independent living, and are performed around 60 times per day in healthy adults [6], enabling functional independence for activities such as walking and toileting. However, transfers (including STS) are the most common cause of falls in people with stroke [7], with STS activities presenting a higher risk of falls for those with impaired strength, balance or posture [8]. Falls are a major hazard for people with stroke, causing injury, lengthening hospitalisations and impacting rehabilitation [9]. Therefore, minimising the risk of falls during these actions is crucial in the creation of assistive devices for stroke rehabilitation.

The ultimate goal of this research is to provide the basis for the development of personalized home assistance for people with stroke during STS transfers, promoting their independence and rehabilitation. Existing manual handling assistive equipment, such as slings and hoists, require constant physical support from the carer to the patient using the device, resulting in increasing work-related musculoskeletal disorders, a major safety concern in today’s healthcare environment [10]. Moreover, the use of a non-perfect fitting device can be uncomfortable or even dangerous, e.g., if it is too large, the patient might slip through.

More advanced robotic solutions have been studied to assist with STS transfer, such as the ballbot [4], a single-wheeled mobile robot that can help a user stand with the aid of two arms that can be held onto during the STS motion. Although the ballbot can apply varying levels of assistance, its maximum pulling force is quite low, and it does not have any functionality for preventing falls, limiting its use with more vulnerable users. Moreover, this robot creates the same profile for all users and is less adaptable to individual needs. Another example is ROBEAR (https://www.riken.jp/en/news_pubs/research_news/pr/2015/20150223_2/, accessed on 26 May 2022), a bear-shaped robot aiding with both STS transfers and rising from bed. ROBEAR, only a prototype, does not have any adaptive capabilities for users of varied ability levels and requires supervision from healthcare workers, limiting its use for maintaining independence. Exoskeletons have also been explored for STS assistance, with some small-scale studies showing that such devices can reduce load on a users’ muscles during STS movements [11,12]. Because these devices are only affixed to the lower body, they benefit from not requiring a user to grip onto handles during STS, but conversely, this creates added risk from the upper body not being supported. Moreover, being “permanently” attached to the participant, exoskeletons may not prove comfortable for everyday use.

One of the main issues with the implementation of these devices into practice is the lack of adaptability. Adaptability in assistive robots is a key enabler, as behaviours can vary between patients who may also progress or regress in their rehabilitation [13]. The assistance provided by the robot should therefore adapt to these individual needs. Failure to do so can result in weakness of the user when too much assistance is provided, or reduced motivation when too little support is given and the person is unable to complete the task. Moreover, an adaptive strategy enhances intuitiveness and natural use of the device, endowing the robot with partial autonomy and unloading the user and/or carer from the burden of regulating the device. To tackle robot adaptability, this project focuses on two main objectives: (i) using biomechanical analysis to understand different motor patterns of sit-to-stand and stand-to-sit movements in individuals with and without stroke, and (ii) trajectory prediction for STS focusing on informing a user-centric adaptive control approach for assistive robots.

To capture full-body biomechanical data, it is possible to use vision-based motion capture systems, such as Microsoft’s Kinect (https://azure.microsoft.com/en-us/services/kinect-dk/, accessed on 26 May 2022). Previous work has used these systems to great effect, particularly when motion is constrained to a single plane [14]. It also shows high reliability for gait analysis [15]. However, vision-based systems present several issues, such as the underestimation of joint angles [16] and larger errors when estimating unconventional positions [17]. Thus, although such systems are lower in cost than IMU-based systems, their potential inaccuracy as well as their often lower capture rate reduces their ability for use in clinical trials. Taking into account these factors, we decided to use more precise IMU-based recordings for the creation of the dataset, while we considered that due to cost-effectiveness and usability issues (people with disabilities might have issues using wearable devices), the developed assistive device would use a machine-vision-based system.

Biomechanical analysis of STS movements has been carried out for many years, with studies focusing on a wide variety of factors often combining adjustments to experimental parameters, such as seat heights [18,19] or allowing arm use for momentum [20]; with measuring biomechanical factors, such as angles and moments of individual joints [18,21] or body weight distribution at points of contact [22]. However, these studies often only capture or provide partial data of the body mechanics. For example, in [21], pressure data were recorded for participants’ feet but not from under the seat, giving an incomplete picture of how weight is distributed and transferred during STS movements. Authors in [19] simplified the human model by negating the participants’ arms for ease of calculation, which then loses balancing and momentum gaining forces generated by the upper limbs. In works such as [19,21,22] among others, the authors enforced strict rules about how the participants should sit and perform the STS movements, which does aid the consistency of findings but fundamentally fails to recreate how people perform STS actions in their everyday lives. Moreover, the data from these studies are usually difficult to retrieve and proprietary, and to the authors’ knowledge, no open-source database containing full biomechanical data for sit-to-stand-to-sit (STSTS) is currently available. Some datasets do exist but are quite limited, such as only using one accelerometer to measure trajectories [23], solely measuring force data from a balance board [24], or using unnatural or ‘perturbed’ motions [25]. Also uncommon are datasets containing data from users with pathological conditions [26]. Another limitation of many existing studies is the focus on sit-to-stand without including stand-to-sit, an equally important transfer. Stand-to-sit may introduce extra risks or complications to the movement trajectory of an impaired person due to the fact that gravity is acting with the participant, requiring additional muscle control from the subject to sit down safely [27]. By capturing the full biomechanical data alongside pressure data on both the seat and floor, for all multiple seat heights, sit-to-stand as well as stand-to-sit movements, and for both stroke and non-stroke participants, this study aims to create a comprehensive open-source dataset available for future research in this field.

Based on these recorded data, a trajectory prediction algorithm can be applied to retrieve the estimated motion of a user. However, though trajectory prediction has been widely documented for walking [26], it is very scarce for STSTS movements. Previous works such as [28], have used cost functions to generate trajectories for different groups of individuals and even applied them to an assistive device achieving high success rates [29]. In [28], full-body motion capture in addition to force data from sensors under the feet and seat were used to calculate trajectories. However, the participants had to stand up from a bicycle seat which would be sat on and stood up from in a different way to a traditional chair; therefore, the trajectories recorded might not necessarily reflect how a participant may move in their everyday life. Similarly to the current study, in [29] an Xsens MVN motion capture suit was used to monitor healthy participants with high accuracy and frame rate for their unimpaired subjects. For the older subjects, however, an optical marker system was used to capture biomechanical data, and this inconsistency may have created extra variations in the data from the two participant groups. Due to the task at hand, where the sample space for participant heights and weights was fairly small, the small distances between datapoints allows algorithms, such as weighted *k*-nearest neighbours (k-NN) [30], to generate accurate results.

The aim of the proposed study in this manuscript is thus to evaluate whether simple machine learning algorithms such as k-NN can be used to predict STSTS trajectories to inform adaptive robotic control strategies for people with stroke.

## 2. Materials and Methods

The methodology for this work is split into two sections. First, the data collection method and the experimental setup for capturing the STSTS data is presented. Second, the processing of the recorded biomechanical data for the prediction of movement trajectory is described.

### 2.1. Data Collection

Participants were recruited from two populations: non-stroke participants recruited from staff and students working within the Bristol Robotics Laboratory, who all reported no physical health problems affecting gait, musculature strength or balance; and those who had experienced stroke. Stroke patients were recruited from Bristol After Stroke, a voluntary organisation in the South West of England. Eligibility criteria were as follows: people with stroke aged over 18 years resulting in unilateral weakness, shoulder muscle strength 3 on Medical Research Council (MRC) scale, medically stable, able to sit to stand independently and able to provide informed written consent. Table 1 shows the distribution of recruited participants from both non-stroke and stroke-affected populations. These criteria were chosen to select participants with mild stroke, or who had significantly recovered from a more serious stroke, and whose STSTS trajectories might not be too dissimilar to those of people without stroke. Analysing mild conditions will (i) reduce the risk of fall and fatigue during the STSTS recordings, and (ii) help validate the proposed methodology in people with mild impairments, after which further research can progress to adapting to users with more severe impairments. We took ‘able to sit to stand independently’ to mean without assistance from mechanical aids or people, as research suggests that later stages of stroke rehabilitation benefit from exercises focusing on lower body strength without external support [31,32].

#### 2.1.1. Experimental Setup

This study combined three synchronised sensors to capture each participant’s STSTS data. Figure 1 shows the setup of these sensors during experiments with a non-stroke subject, and Figure 2 outlines the experimental procedure.

A full-body motion capture suit (MVN Link, Xsens Technologies, Enschede, The Netherlands) (https://www.xsens.com/, accessed on 26 May 2022), with 17 Inertial Measurement Units (IMU) held tightly to the body with rubberised straps. These were positioned on each shoulder, upper arm, wrist, hand, thigh, shin and foot, as well as the pelvis, sternum and head. Each sensor transmitted position and orientation at 100 Hz. Sensors on the feet were attached using tape to better maintain a fixed position. The XSens sensors are wired together in a chain to a power pack and transmitter device located on the participant’s back, with enough slack to allow freedom of movement. The data are streamed from the transmitter to a PC wirelessly.

A pressure sensing mat (Seating Mat Dev Kit, SensingTex, Barcelona, Spain) (http://sensingtex.com/sensing-mats/seating-mat/, accessed on 26 May 2022). The seating mat comprised 16 × 16 pressure sensors in a 30 × 30 cm square fabric, placed with the participant sitting in the centre. Each sensor output an integer value from 0 to 1024, at 4±1 Hz. This enabled analysis of a participant’s weight distribution when seated and preparing to stand and when returning to the seated position.

Finally, a bespoke balance board, which consisted of a square plywood sheet with force sensors (Model YZC-161B) in the four corners in a 0.855 × 0.86 m grid to measure a participant’s centre of force in the X-Y plane, transmitted at 85±5 Hz.

Each of these sensory systems had its own calibration procedure. The XSens suit calibration was built into the XSens MVN analyze software (https://www.xsens.com/products/mvn-analyze, accessed on 26 May 2022), and consisted of a routine where the participant had to sequentially stand statically with feet hip width apart, arms straight and head forward for 5 s; walk forwards at a comfortable pace for 5 s; and turn 180 degrees and walk back to their starting position. For the SensingTex Seating Mat, an attached microcontroller unit contains firmware that translates the pressure signals from each sensor into 8-bit values. This was then calibrated by adding known weights to the seating mat and converting the total reading from all sensors under pressure into a weight value. The bespoke balance board contains a an Arduino Nano (https://store.arduino.cc/products/arduino-nano, accessed on 26 May 2022) microcontroller which converts the force sensor signals into weight values. This was calibrated by adding known weights to the individual sensors and adjusting internal variables to obtain accurate readings.

#### 2.1.2. Experimental Protocol

Previous research has suggested standard methods for measuring STS biomechanics. Often, the sit-to-stand motion is repeated 3–5 times per trial, with breaks in between trials to avoid patient fatigue. Seats can be set to a range of fixed heights [19,21], or increased as a percentage of knee height for each participant [33,34,35]. Participants are often asked to keep their arms crossed in from of their chest [21,34]. Instead, in this study, we allowed the motion of the arms to (i) record natural movements, and (ii) reduce the risk of falls introduced by monitoring people with stroke who may suffer from impaired balance. Participants were asked to keep their hands in a comfortable position, but to avoid contacting external surfaces to provide extra leverage during the STSTS trials. Research has shown that a proportion of elderly and impaired users prefer to use external surfaces, such as chair arms or their own knees, as compensatory behaviours [36], but in contrast, some stroke rehabilitation trials have chosen to prohibit the use of these techniques, resulting in improved strength and function of the lower limbs [31,32], helping to support the reason for using these limitations in the current study.

In this study, each participant performed four sets of 5 STSTS actions, completing a total of 20 repetitions. Based also on the literature regarding STS experiments with people with stroke [8], we decided to perform the first two sets using a seat height of 100% of the participant’s knee height (measured from the floor to the base of the patella), and the latter two using 115% knee height. In some previous studies, lower seat heights, such as 60% or 85% knee height, were also tested, which were found to increase forces required on the knee and hip joints in order to stand. These lowered seat heights were not examined in this study to avoid extra strain on participants during sit-to-stand, and less velocity during stand-to-sit actions. Participants were asked to complete the 5 STSTS at their own pace, raising a hand above the shoulder as a marker of completion of the action. Participants were allowed to take as much time as required between sets to allow for full recovery.

### 2.2. STSTS Trajectory Prediction

Figure 3 outlines the data processing used to obtain final movement trajectories. Trajectories were predicted using only data from the non-stroke participants, so that generated trajectories would be based on ‘ideal’ data and not influenced by impaired movements.

Data from the seating mat and balance board were synchronised using interpolation to match the 100 Hz of the XSens suit and create identical timestamps. To do so, the Python’s SciPy library [37] was used with the linear interpolation function. Recordings of each set of 5 STSTS actions were split into individual sit-to-stand and stand-to-sit trials using the participant’s hand raise as a marker. The data from each action were manually trimmed to remove excess data from the onset of the actual movement to the end. This was done to ensure that trial averages were only calculated on data from the participants actual movement, and was based on the neck position on the assumption that the upper body is first to move during STS movements to gain momentum.

To obtain average trajectories for each participant, following previous research [11,29], the 10 trials for each seat height were first stretched to equal length (n=1000 frames) by interpolation, after which the mean and standard deviation for the trials could be calculated. After each user’s average trial had been calculated, the next stage was to begin creating a model to match these average trials based on users of similar height and weight, or age and Body Mass Index (BMI).

The literature often highlights how displacements of the centre of mass (CoM) are a predictor and cause of falls [7,22]. Calculating CoM dynamically is computationally expensive, and many studies opt for using simplified human models [38], which has been show to affect the accuracy of resultant predictions [39]. This study focuses on the potential use of robotic assistance to prevent users arriving at an unstable body position that could lead to a fall. Taking into account that shoulders are a commonly included key points for motion tracking using computer vision systems [40], this study opted for using an easy-to-estimate virtual point in between the shoulders of each participant as the origin for the created trajectory. These virtual data were created using the midpoint of the Left ($Z_{sl}$) and Right ($Z_{sr}$) shoulder positions recorded from the Xsens data, calculated using Equation (Equation 1).
(1)$$Z_{m}=\frac{Z_{sr}+Z_{sl}}{2}$$

Afterwards, we focused on obtaining a simplified prediction algorithm that needs as few variables as possible to retrieve the user’s trajectory and can be seamlessly implemented in a robotic solution. We focused on the sagittal plane of movement (*x* axis in Figure 1), assuming that robotic support will be predominantly in the vertical direction providing assistance for weight bearing, and the robot would behave as a follower in the longitudinal direction. Nevertheless, the movement and behaviour in all three axes were captured and are published to enable further research in this area (see Section 3).

To predict the trajectory, the weighted k-nearest neighbours (k-NN) algorithm [30] was implemented using the coordinates of participant height against participant weight, as well as for age against BMI using Equation (Equation 2). Each participant was removed from the dataset in turn, with their height and weight or age and BMI values reserved as a coordinate for test data. The nearest neighbours to this test coordinate were then weighted according to their distance to the test coordinate, then combined to create a single trajectory at each time stamp *i*, $Z_{i}$. Additionally, we assumed the possibility of the user manually adjusting an assistive device to a comfortable height before performing a sit-to-stand action. Thus, the true starting measured position $Z_{m,0}$ for the removed user was saved, and the predicted mid-shoulder trajectory at each time instant *i* shifted to a new estimate $Z_{i}^{\prime }$, matching this true starting point.
(2)$$BMI=\frac{Weight(kg)}{Height{\left(m\right)}^{2}}$$
(3)$$Z_{i}^{\prime }=Z_{i}-Z_{0}+Z_{m,0}$$

The height and trajectory end points for all users (except the removed test user) were then plotted against each other, producing a weak positive correlation (0.28 using python’s SciKit Learn library [41]). A linear regression model was then built off these data [42], which, when fed the test user’s height as an input, could predict their end point $Z_{r,n}^{\prime }$, where *n* is the length of the trajectory vector. The predicted trajectory was then scaled to finish on this predicted end point.
(4)$${Z^{\prime \prime }}_{i}=Z_{i}^{\prime }+\frac{i}{n}\left(Z_{r,n}^{\prime }-Z_{n}^{\prime }\right)$$

The true trajectory for the removed participant was compared with this generated trajectory, and the coefficient of determination $R^{2}$ [43] for these two curves was found using Equation (Equation 5). This was repeated for each user, and then for different values of *k* (*k* = 2, *k* = 3, *k* = 4, *k* = 5). The average and standard deviation of the $R^{2}$ value over all users for each value of *k* were calculated. Terms are as described previously, and ${\bar{Z}}_{i}$ represents the mean value of *Z* at time step *i*.
(5)$$R^{2}=1-\frac{\sum_{i=0}^{n}{\left(Z_{i}-{Z^{\prime \prime }}_{i}\right)}^{2}}{\sum_{i=0}^{n}{\left(Z_{i}-{\bar{Z}}_{i}\right)}^{2}}$$

Pressure data from the four balance board sensors were analysed by observing the difference in pressure on each side of the board to determine how participants weighted each of their legs during the STSTS movements:(6)$$\begin{array}{c} P_{L}=P_{FL}+P_{RL}\\ P_{R}=P_{FR}+P_{RR}, \end{array}$$
with $P_{FL}$ and $P_{FR}$ representing the pressure on the front left and right sensors respectively, $P_{RL}$ and $P_{RR}$ representing the pressure on the rear left and right sensors, respectively.

The centre of pressure (CoP) was also retrieved from balance board data (Equation 7):(7)$$\begin{array}{c} CoP_{x}=\frac{P_{FL}\;FL_{x}+P_{FR}\;FR_{x}+P_{RL}\;RL_{x}+P_{RR}\;RR_{x}}{P_{FL}+P_{FR}+P_{RL}+P_{RR}}\\ CoP_{y}=\frac{P_{FL}\;FL_{y}+P_{FR}\;FR_{y}+P_{RL}\;RL_{y}+P_{RR}\;RR_{y}}{P_{FL}+P_{FR}+P_{RL}+P_{RR}} \end{array}$$

$FL_{x}$, $FL_{y}$, $FR_{x}$, $FR_{y}$ represent the *x* and *y* distance of the front left and right sensors from the centre of the balance board, and $RL_{x}$, $RL_{y}$, $RR_{x}$, $RR_{y}$ represent the *x* and *y* distance of the rear left and right sensors from the centre of the balance board, with the exact measurements shown in Equation (Equation 8) (values in *m*).
(8)$$\begin{array}{c} \left(FL_{x},FL_{y}\right)=\left(0,0.855\right)\\ \left(FR_{x},FR_{y}\right)=\left(0.86,0.055\right)\\ \left(RL_{x},RL_{y}\right)=\left(0,0\right)\\ \left(RR_{x},RR_{y}\right)=\left(0.86,0\right) \end{array}$$

We hypothesise that people with stroke, who may have impaired balance, would find it harder to regulate their centre of pressure during STSTS movements, which would be shown by a larger range of CoP values for these participants.

## 3. Results

The results from this study are split into two sections. First, we present the analysis of the data captured from the seating mat and balance board sensors. These are not used later for trajectory generation, but provide a valuable qualitative comparison of the non-stroke participants against those with stroke. Second, we validate the k-NN and linear regression algorithms on predicting STSTS trajectories using the data from non-stroke participants using the XSens motion capture suit.

All the presented and used data in this section are publicly available in both raw and processed forms at [44].

### 3.1. STSTS Dataset Analysis

#### 3.1.1. Balance Board

Figure 4a shows the average force applied to the balance board’s left sensors (front and rear) compared with the force on right sensors for selected non-stroke and stroke participants over 10 repetitions with seat set at 100% knee height. For the non-stroke participants, a similar amount of force was applied on each side of the body during the sit-to-stand motions, whereas for the stroke-affected participants, a lot more of the weight was taken up by the unimpaired side, in this case, the left leg. This is particularly noticeable around the middle of the STS motion, as weight is being transferred from the seat and onto the feet but before the participant has become fully stable. Also noticeable for the stroke participants is a reduction in force on the balance board just after the beginning of the motion.

Figure 4b shows similar features for the stand-to-sit movement, with the stroke subject greatly favouring the unimpaired leg over the impaired, and the non-stroke subject showing much more even weighting of the legs. The mean difference in weight applied to each leg over the whole action was calculated for each participant using Equation (Equation 9), where $P_{FL,i}$ represents pressure on the front left force cell at time step *i*. The average difference for non-stroke subjects was 2.08 ± 1.75 kg, and for stroke subjects, this was 4.53 ± 4.05 kg. For stand-to-sit, these values were 2.52 ± 1.88 kg for non-stroke and 5.13 ± 4.07 kg for stroke participants.
(9)$$Weight\;Difference=\frac{\sum_{i=0}^{i=n}\left|\left(P_{FL,i}+P_{RL,i}\right)-\left(P_{FR,i}+P_{RR,i}\right)\right|}{n}$$

Figure 5 shows the differences between centre of pressure trajectories for stroke and non-stroke participants. Some of the stroke-affected participants show much larger displacements in both the *x* and *y* directions, demonstrating that these participants struggled with balance, and may be at greater risk of falls. Non-stroke participants generally showed little displacement in the *x*-direction.

#### 3.1.2. Seating Mat

Though the full quantitative analysis of the seating mat data is out of the scope of this paper, here we present some brief qualitative analysis highlighting the differences between non-stroke and stroke participants. Figure 6a shows the average progression of pressure applied to the seating mat over the course of 10 sit-to-stand movements for two representative participants, one non-stroke, and one with stroke. Each coloured pixel represents the weight in kilograms applied to each sensor on the mat. It must be noted that this image does not show the full STS trajectory, as when the participant is preparing to stand, and when their weight is fully off the mat, the pressure on the seating mat remains quasi-static.

It can be observed that for the non-stroke subject, the seat-off movement was more gradual and smooth, taking around 40% (55% − 15%) of the total movement time to reach seat off. Instead, the stroke participant created more instant velocity to generate the necessary impulse for seat-off, thus shortening the time of this event (around 25% of the total movement). Also notable is that, as can be observed in Figure 6a, the stroke participant was seated further forward on the seating mat, having less contact surface with the chair.

Figure 6b shows the average weight distribution on the seating mat for the same stroke and non-stroke subjects, this time during the stand-to-sit action. In this graph, matching timestamps are shown for each subject, and it is clear again that the participant with stroke greatly favours one leg (in this case their right) for weighting first, and only reduces weight on that leg at the very end of the action when they have achieved a stable position. Again, the stroke subject sits further forward on the seating mat. The non-stroke participant shows a slight favouritism of one limb over the other, but much less pronounced than the non-stroke user, and weights both limbs at a similar rate.

### 3.2. STSTS Trajectory Prediction

Figure 7 shows a visualisation of the marker data captured by the XSens suit, overlaid onto images of a non-stroke participant performing the sit-to-stand motion. This image shows the progression of the participant over key moments of the sit-to-stand movement, including seated stationary, preparing to stand, seat off, obtaining standing balance, and finally standing stationary.

Table 2 and Table 3 show the average $R^{2}$ values for different *k*-nearest neighbours when comparing each non-stroke participant’s true trajectory against that predicted by the algorithm. Table 2 shows results when using height and weight as k-NN coordinates, while Table 3 shows results when using age and BMI. The highest score is for k=3 for height and weight coordinates, with the average $R^{2}$ score over all participants of 0.864 ± 0.134 for sit-to-stand at 100% seat height.

For both types of k-NN coordinates, sit-to-stand gave higher average $R^{2}$ results than stand-to-sit, and also 100% seat height gave higher results than 115% seat height, as well as larger standard deviations in both cases. Using height and weight gave very similar results for sit-to-stand than age and BMI, with a maximum difference of 0.036 for k=2 at 115% seat height. However, age and BMI coordinates did generate more accurate trajectories for stand-to-sit, particularly for higher values of *k*. The maximum difference was 0.173 for k=4 at 115% seat height, also showing a standard deviation almost half the size of that for height and weight coordinates.

Only 6 stroke-affected participants were measured, each with significantly varying levels of impairment and movement patterns. Due to the small and diverse sample, averaging stroke participants data would have neglected the differences that arise from having suffered from impairments on different sides of their bodies. Thus, we made the decision to display and analyse each participant with stroke separately rather than focusing solely on average values.

Figure 8 shows each stroke participants’ sit-to-stand and stand-to-sit graphs, at 100% seat height on the left and 115% seat height on the right. Red lines show the true trajectories for each participant, averaged over their 10 trials, with standard deviation lines also shown. Blue and green lines represent trajectories predicted for that user by the k-NN and linear regression model, using k=3. These graphs show a huge variation in how well the predicted trajectory matches with the participants’ true movement, with a top $R^{2}$ score of over 0.99 for the third participant’s sit-to-stand action at 100% seat height, but only 0.11 for the same action for the first participant. The average $R^{2}$ score for the sit-to-stand action at 100% seat height for the stroke participants was 0.816±0.267 when using age and BMI as k-NN coordinates, which is a slightly lower average than that for the non-stroke participants, and with a much higher standard deviation. Sit-to-stand movements at 115% seat height are closest in average score to those from the non-stroke participants, with even a smaller standard deviation. Table 4 and Table 5 show the full results for each participant and action. Note that in this case, the predicted trajectory represents an ‘ideal’ behaviour of the participant as a non-stroke subject. A larger error (smaller $R^{2}$ value) corresponds to a larger deviation between the stroke and ‘ideal’ trajectories.

## 4. Discussion

This study used a k-NN and linear regression model to produce trajectories for unseen participants during sit-to-stand and stand-to-sit movements, which in most cases only deviated from the true path by a few centimetres (see Table 2 and Table 3). This was most accurate for subjects near to average heights and weights, as their nearest neighbours were more likely to be a lot closer than for subjects on the fringes of the dataset. This model was able to recreate the full mid-shoulder trajectories from non-stroke subjects with an average $R^{2}$ score of 0.864 ± 0.134 for sit-to-stand movements starting from 100% seat height and using height and weight as k-NN coordinates, in most cases deviating from the true trajectory by no more than 5 cm. For 115% seat height, though we still had achieved some level of prediction, the $R^{2}$ score decreased and error increased for both sit-to-stand and stand-to-sit movements. This could be due to a raised seat height being less familiar to participants, which as a result created more variation in how they distributed their weight and generated momentum, changing their STS patterns. Additionally, balance could have played a role in the repeatability of the motions, as at 115% height, the seating position was less stable and the impulse to stand up could have been less predictable. A larger number of participants would increase the accuracy of the model, and is an aim for the future of this project.

Comparing the use of height and weight, and age and BMI as k-NN coordinates, the former provided the highest single score, but only by a very narrow margin, whereas age and BMI was shown to more successful for stand-to-sit trials, including having a generalized smaller error. Age and BMI was also more successful at predicting trajectories for higher values of *k*, which can avoid chances of overfitting [45]. For these reasons, our results suggest that using age and BMI as k-NN parameters is more successful for predicting STSTS movements.

After validating the prediction model, it was also used to estimate ‘ideal’ trajectories for stroke subjects, which could be used for the control of an assistive devices. As it can be observed in Figure 8 and Table 4 and Table 5, the difference between ‘ideal’ and real trajectories varied depending on the subject. For example, subject S1 exhibited a deep lowering behaviour at around 40% of the movement, that reflected the impulse taken during the sit-to-stand and that is not present in an ‘ideal’ trajectory. Thus, a control could be envisioned in which the robot tries to follow such ‘ideal’ trajectory depending on the level of impairment and assistance needed by the subject. Conversely, participant S3 was in a good physical state and had little level of impairment, which was visible from the small difference between the the ‘ideal’ and measured trajectories. Thus, in this case, the robot would provide little assistance to the person’s movement. Table 4 and Table 5 shows that using age and BMI as k-NN coordinates results in higher average $R^{2}$ values across all movements, and therefore is more appropriate for use in future work.

Moreover, in this work, we managed to create a comprehensive data-set of full-body biomechanics and force data for 30 participants, including 6 who had experienced stroke. These data allowed for a first comparison analysis of body motion for people with and without stroke during STSTS movements, and opens the door for further research on physiological studies and robotic assistance.

Based on the results shown in Figure 5, the range of displacement during sit-to-stand, particularly in the *x*-direction of the CoP, could be used as an indicator of potential falls, and utilized by an assistive robot to take preventive actions. Despite being able to see large differences in body position and applied forces (Figure 8) for some stroke subjects, the small sample size of stroke participants meant that quantitative analysis of these differences could not be performed reliably. However, the large range of ability levels within the stroke subjects group highlights the importance of future assistive devices to be adaptive to the individual user, and thus to have personalized predictions of their movements.

It must be also pointed out the difference in ages between the two groups, with averages of 37.2 ± 12 years for non-stroke and 66.5 ± 10.7 years for stroke participants. Although some non-stroke participants tested were in their fifties and sixties, the majority were much younger than participants in the stroke group. This could mean that some of the biomechanical differences seen in the stroke group were influenced by age as well as the effects of the stroke. Further research will expand the number of stroke participants tested, including younger people with stroke, to reduce this age discrepancy.

Other trends noted in the stroke-affected participants included a ‘rocking back’ motion when preparing to stand, seen both in the seating mat data with a higher weight applied just before standing, Figure 6, and also mirrored in the balance board data, Figure 4, where participants removed some weight from the feet just before acting. Additionally, some stroke participants exhibited a large difference between weight applied to each foot during STSTS motions, highlighting favouritism of their unimpaired limb and potential increased risk of falls. A key observation is that these trends were not homogeneous among the groups, with some stroke participants showing very little difference between weight applied to each foot, with some non-stroke participants also showing some favouritism of one leg. Larger sample sizes would help to determine more definitive trends within these groups, and create segmentation within groups further than just ‘stroke’ and ‘non-stroke’.

It is possible that some of the stroke-affected participants may use compensatory behaviours, such as pushing off the arms of a chair when performing STS movements in their everyday lives [36,46]. Prohibiting these behaviours in this study may have affected how the participants performed STSTS movements, but care was taken to ensure that all participants were able to safely perform STSTS movements independently.

People with stroke demonstrate considerable asymmetry of weight distribution during sit to stand, demonstrating significantly increased weight bearing on the unaffected side [47,48]. Stroke survivors also commonly exhibit a reduced peak vertical reaction force, an increased time to complete the movement of sit to stand and a larger medio-lateral centre of pressure displacement compared with healthy adults [47]. This also can be seen when looking at the data from the force sensors in Figure 4; the non-stroke participant spreads the weight much more evenly over their two legs, and it is clear that they favour weighting one leg throughout the whole movement, which is also shown in the higher weight distribution imbalance for stroke participants using Equation (Equation 9). The non-stroke person sits much further back on the seating mat than the person with stroke in Figure 6, and it has been shown that sitting or ‘scooting’ forward on a chair can make performing STS movements easier for elderly people [46,49], thus showing that the non-stroke user is able to apply more torque to their legs. Comparing the first frame of the stroke patient with the second and third frames, more weight is applied to the seating mat just before the patient stands, representing the patient ‘rocking’ backwards to build momentum for the stand movement. This was also observed visually during the trials.

The data created in this research are a valuable starting point for further designs of assistive robots. We demonstrated that simple machine learning algorithms are able to predict STSTS movement trajectories with relative accuracy, which could be used as inputs for a robot aiming to follow a user’s STSTS trajectory. We highlighted areas where true trajectories differ from predictions, which a robot could use to adapt the level of assistance it is providing to guide the user back to a safe path. We showed that impaired participants tend to exhibit more unstable forces when standing, which could be used as indicators of potential falls and trigger a robot to take preventative measures. Building upon previous datasets found in literature [23,24,25], this dataset combines full-body biomechanics with force data from multiple sensors, allowing for more thorough analysis and comparison between sensors. Further exploration of this dataset could involve analysing participant velocities and accelerations to determine the maximum forces that a STSTS assistive robot would need to be able to apply to operate safely for a wide variety of users. Another avenue could be to focus on a participants’ movement in the horizontal plane and their centre of pressure, as a robot may attempt to prevent falls by reducing movement in these parameters.

A limitation of this study is that it focused on stroke participants who were able to stand safely by themselves to minimize the risk of falls or injury. Stroke participants were recruited from within the Bristol After Stroke charity, which is based in the community rather than in hospital or other rehabilitation settings, and therefore supports more able patients. This means the dataset does not include trajectories of persons severely physically impaired by stroke and therefore does not show the likely larger variation in trajectories, body positions and applied forces that these subjects might demonstrate.

However, a STSTS assistive robot is likely to target a wider range of impaired subjects, possibly including those who might be unable to stand independently. Further research should focus on testing and recording these user’s biomechanical data when given the minimum assistance required to allow them to perform STSTS movements. These recordings would help to determine physical additional requirements of a STSTS robot, such as workspace, force limits, and safety measures, as more severely impaired users are likely to exhibit more extreme behaviours. Measuring this variety of movement patterns will be useful for the safety testing and validation of such robots. In addition to a wider variety of user ability levels, future work could investigate more variability in STSTS scenarios, such as rising from lowered seats, which, although are not ideally suited for impaired users, are still situations that may arise in a home environment.

## 5. Conclusions

This manuscript presents an algorithm to estimate sit-to-stand and stand-to-sit trajectories from a minimal set of biomechanical parameters. The proposed model was able to recreate full mid-shoulder trajectories from non-stroke subjects with an average $R^{2}$ score of 0.864 ± 0.134. The model was then used to predict ‘ideal’ trajectories for people who have suffered a stroke. Moreover, pressure data from both seat and underfoot sensors were presented alongside full biomechanical recordings to form an open-source database (Section 3) that will serve to inform future studies in STSTS motion assistance. The predicted trajectories could be used by an assistive device to guide the motion of a person who has experienced a stroke towards a pre-stroke pattern. The lack of previously existing comprehensive datasets accentuates the significance of the presented open-source data, which can enable further analysis to be carried out and combined with the findings in this study. Additionally, the use of simple machine learning techniques to predict personalized trajectories with minimal biometrical inputs, shows that future assistive devices do not necessarily need to rely on complicated systems to provide adaptable assistance to different users. Future work will focus on applying the generated trajectories into an existing assistive robot, to assess the safety and comfort of the generated control.

## Figures and Tables

**Figure 1 sensors-22-04789-f001:**
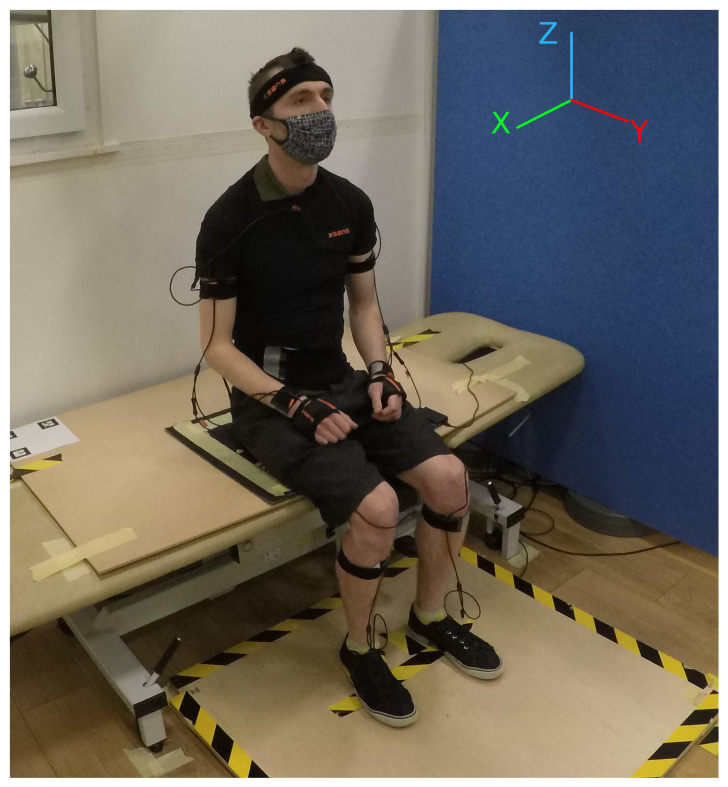
Participant wearing XSens suit, seated on the sensor mat attached to a rigid board on an adjustable plinth, with feet placed on the balance board. This setup shows the seat at 115% knee height. Reference frames are shown for guidance on the following methods and results. Yellow and black tape in the middle of the balance board was to help the participants stand near the centre, although they were allowed to move their feet at will to keep themselves balanced and comfortable.

**Figure 2 sensors-22-04789-f002:**
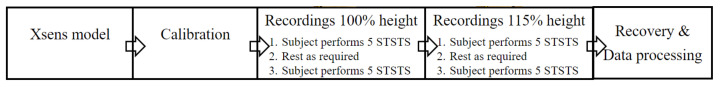
Flow chart depicting data capture procedure for each participant.

**Figure 3 sensors-22-04789-f003:**
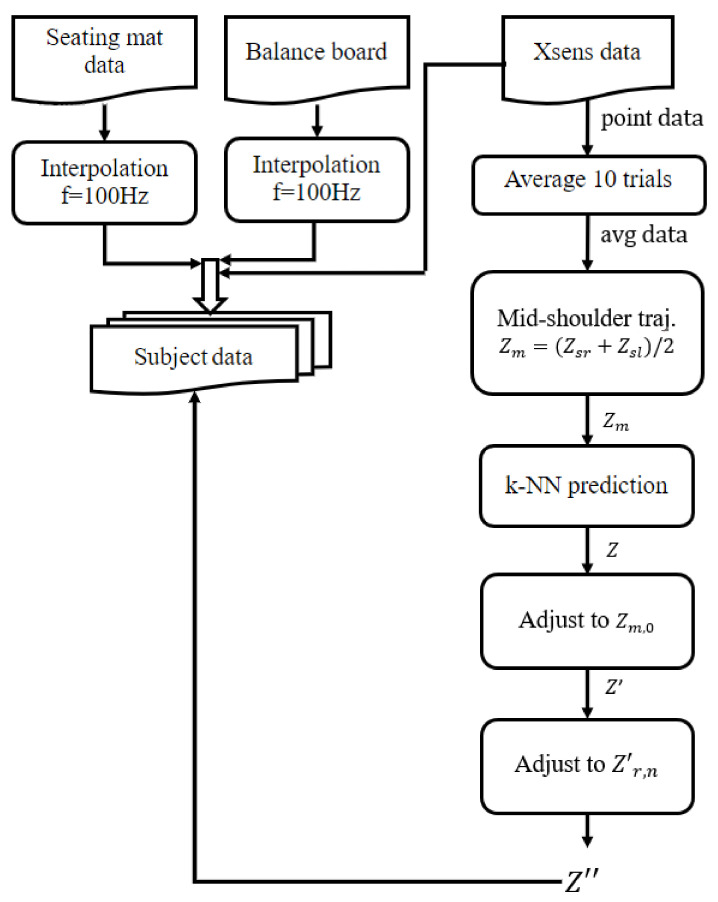
Flow chart depicting data processing and trajectory prediction for each participant.

**Figure 4 sensors-22-04789-f004:**
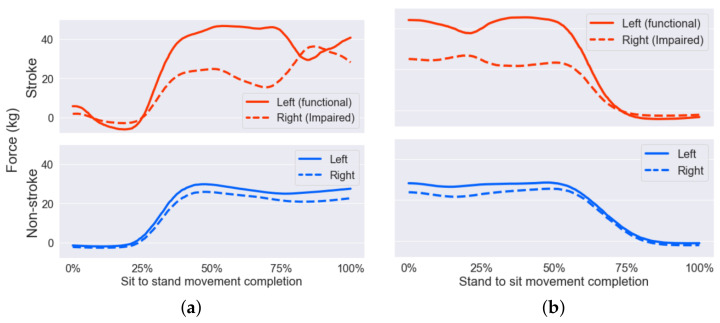
(**a**) Average weight placed on each side of the balance board for stroke (**top**) and non-stroke (**bottom**) users, 10 sit-to-stand movements, at 100% seat height. (**b**) Weight placed on each side of the balance board for stroke (**top**) and non-stroke (**bottom**) users, stand-to-sit movement, at 100% seat height.

**Figure 5 sensors-22-04789-f005:**
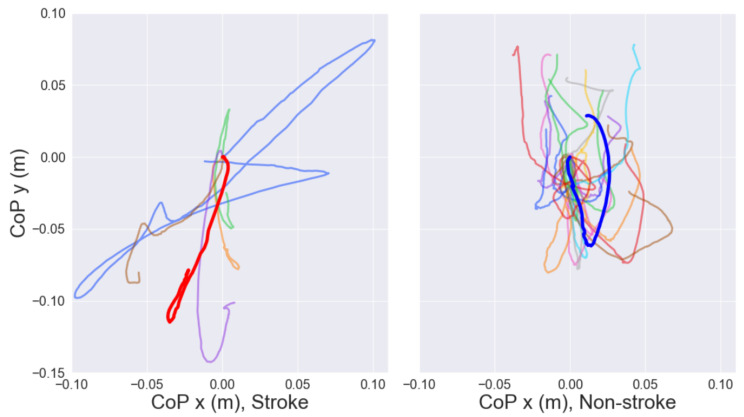
Centre of pressure trajectories for stroke and non-stroke participants, with the start position of each line normalised to (0,0). Each colour represents a participant. Single examples highlighted solely for clarity of comparison.

**Figure 6 sensors-22-04789-f006:**
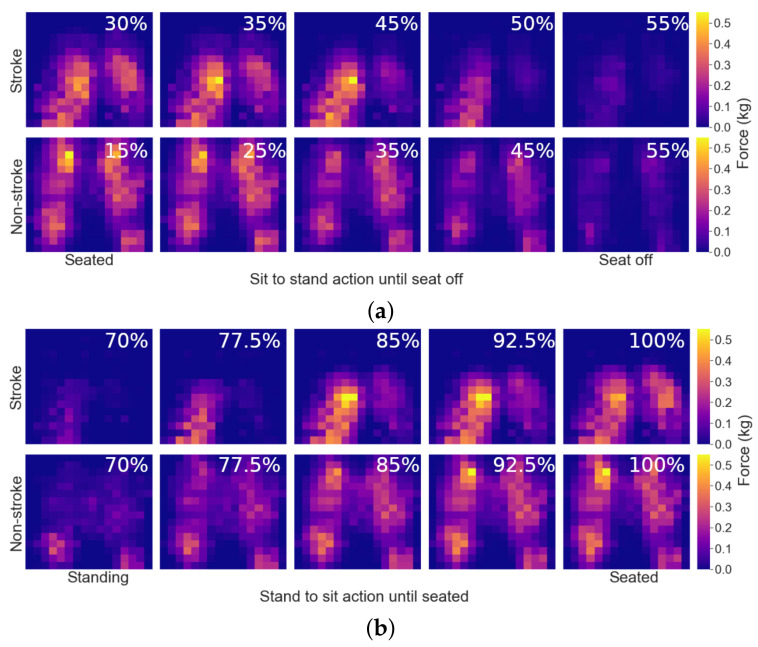
Average weight distribution on seat mat during (**a**) sit-to-stand action, and (**b**) stand-to-sit actions. Each sensor reading weight is in kg. Percentage progress through movement is highlighted in white.

**Figure 7 sensors-22-04789-f007:**
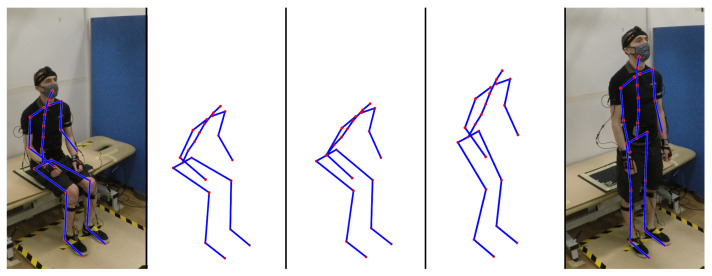
Example of full sit-to-stand action, showing points captured from individual markers.

**Figure 8 sensors-22-04789-f008:**
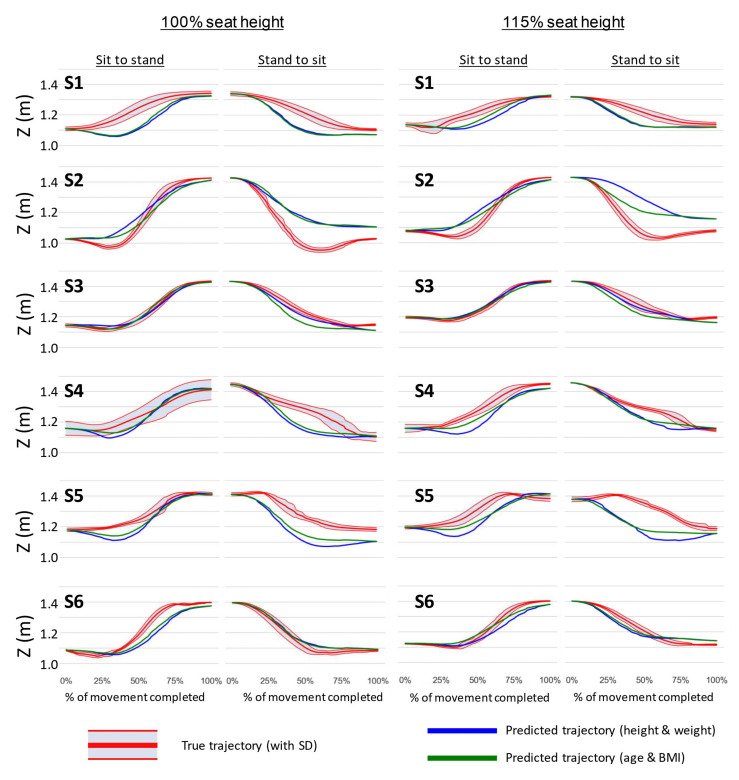
Trajectories predicted by k-NN and linear regression model imposed over stroke participants’ (labelled **S1**–**S6**) recorded trajectories. Two left columns show sit-to-stand and stand-to-sit for 100% seat height. Two right columns show sit-to-stand and stand-to-sit for 115% seat height. Red lines show participants average true trajectory with standard deviations. Blue lines are trajectories predicted by the k-NN and linear regression model using height and weight as k-NN coordinates. Green lines are for predicted trajectories using age and BMI as k-NN coordinates. The *y* axes on each graph represent the *Z* position of the mid-shoulder point, and the *x* axes show percentage completion of the STSTS movement.

**Table 1 sensors-22-04789-t001:** Parameters with means and standard deviations of participants.

	Non-Stroke, *n* = 24	Stroke, *n* = 6
Gender Split (M/F)	14/10	3/3
Age (years)	37.2 (±12.0)	66.5 (±10.7)
Height (cm)	175 (±8)	170 (±5.3)
Weight (kg)	74.7 (±14.9)	87.0 (±18.0)

**Table 2 sensors-22-04789-t002:** Average $R^{2}$ with standard deviation values for different values of *k* when using height and weight as k-NN coordinates.

	100% Seat Height		115% Seat Height	
*k*	Sit-to-Stand	Stand-to-Sit	Sit-to-Stand	Stand-to-Sit
2	0.854 ± 0.138	0.666 ± 0.448	0.719 ± 0.357	0.639 ± 0.397
3	0.864 ± 0.134	0.653 ± 0.376	0.762 ± 0.323	0.579 ± 0.443
4	0.832 ± 0.186	0.516 ± 0.570	0.784 ± 0.281	0.441 ± 0.644
5	0.830 ± 0.215	0.617 ± 0.453	0.799 ± 0.247	0.552 ± 0.543

**Table 3 sensors-22-04789-t003:** Average $R^{2}$ with standard deviation values for different values of *k* when using age and BMI as k-NN coordinates.

	100% Seat Height		115% Seat Height	
*k*	Sit-to-Stand	Stand-to-Sit	Sit-to-Stand	Stand-to-Sit
2	0.861 ± 0.152	0.645 ± 0.316	0.755 ± 0.324	0.598 ± 0.356
3	0.854 ± 0.151	0.723 ± 0.261	0.754 ± 0.324	0.676 ± 0.314
4	0.833 ± 0.186	0.703 ± 0.294	0.759 ± 0.284	0.614 ± 0.358
5	0.852 ± 0.196	0.733 ± 0.266	0.787 ± 0.273	0.635 ± 0.332

**Table 4 sensors-22-04789-t004:** $R^{2}$ values for different stroke participants for sit-to-stand and stand-to-sit actions at different seat heights, using k=3 and height and weight as k-NN coordinates.

Participant	Sit-to-Stand, 100%	Stand-to-Sit, 100%	Sit-to-Stand, 115%	Stand-to-Sit, 115%
S1	0.112	0.372	0.495	0.287
S2	0.929	0.474	0.894	−0.007
S3	0.989	0.966	0.991	0.967
S4	0.843	0.387	0.674	0.759
S5	0.708	−0.509	0.365	−1.82
S6	0.823	0.966	0.917	0.9
**Average**	0.734±0.320	0.443±0.541	0.723±0.253	0.181±1.051

**Table 5 sensors-22-04789-t005:** $R^{2}$ values for different stroke participants for sit-to-stand and stand-to-sit actions at different seat heights, using k=3 and age and BMI as k-NN coordinates.

Participant	Sit-to-Stand, 100%	Stand-to-Sit, 100%	Sit-to-Stand, 115%	Stand-to-Sit, 115%
S1	0.282	0.281	0.752	0.355
S2	0.952	0.482	0.920	0.488
S3	0.996	0.819	0.986	0.832
S4	0.929	0.584	0.862	0.825
S5	0.845	0.137	0.638	−1.065
S6	0.893	0.972	0.963	0.932
**Average**	0.816±0.267	0.546±0.316	0.854±0.135	0.395±0.749

## Data Availability

Data available in a publicly accessible repository at https://github.com/Assistive-Robotics-Lab/STS_Dataset [44], accessed on 26 May 2022.

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
