# Peer review of "A Machine Learning Model for Predicting Sit-to-Stand Trajectories of People with and without Stroke: Towards Adaptive Robotic Assistance"

_sensors, 2022, doi:10.3390/s22134789_

Round 1

Reviewer 1 Report

The paper present a machine learning approach for a database of full body biomechanics and force data captured during sit-to-stand movements for patients with and wthou storke. 

The algorithm to estimate sit-to-stand and stand-to-sit trajectory was tested and s then used to predict ‘ideal’ trajectories for people who have suffered  a stroke. Moreover, pressure data from both seat and underfoot sensors was presented.

The paper is interesting and present a innovativa database that can be used in the emerging field of assistive robotics.

Norever some more key point have to be addressed referencing to this papers that uses less complex ways using just computer vision. Pointing out why all the wear procedure is needed for patient soffering for tdisorders. What advantajes in biomechanics come merging all this metodologyies for skeleton tracking while this can just be done using only computer vision. Plese add this references using these methods and point out advangeges and disavantages of your procedure. this will strenghten the soundness of the proosed paper:

1. https://doi.org/10.1155/2018/9303282

2. https://doi.org/10.3390/app10175781 

Please detail more the inclusion critiria giving a bit more of motivation why such chactristcs have been chosen

Please improve figure 6, it hard to understand.

The paper is of great quality and after this minir revision detailing some parts it is very valuable for pubblication.

Reviewer 2 Report

Sit-to-stand transfers are fundamental daily motions that enable all other types of ambulation and gait. However, the ability to perform these motions can be severely impaired by different factors such as the occurrence of a stroke, limiting the ability to engage with other daily activities.

This authors  presents a study based on a database of full body biomechanics and force data captured during sit-to-stand movements in subjects who have and have not experienced stroke.

The data provided by authors is then used in conjunction with simple machine learning algorithms to predict vertical motion trajectories that could be further employed for the control of an assistive robot.

A total of 30 people (including 6 with stroke) we included in the design and a proper methodology was applied..

The authors made publicly the full dataset of sit-to-stand and stand-to-sit actions for each user for further research.

The manuscript is interesting ad adds an important contribution to the literature.

These are my comments for the improvement with a pure  academic spirit.

1- The abstract must contain a summary of all the sections.

2.     The aim is clear however it could be better enlarged and explicated to highlight the significance of the manuscript.

3.     In the method a flow chart could improve the clarity  of the design

4.     The limitations ends with “For a STS assistive robot, however, more impaired subjects would be a feasible audience. Further research could focus on testing and recording these participants” Try to better expand this clarifying how to enlarge the further research.

5.     Conclusions usually does not contain references

6.     Add a table with acronyms

Reviewer 3 Report

The authors describe machine learning algorithms to predict vertical mid-shoulder trajectories that could be further employed for the control of an assistive robot during sit to stand to sit motions. In my opinion, the manuscript lacks clarity and methodological aspects of the protocol are not clearly described and may not be appropriate. In more detail:

11)     Line 96, “and up to the authors’ knowledge no open-source data base is currently available.” I would disagree with this statement given the very limiting number of stroke patients in this dataset and the limitations in the data gathering process (please also see comments below).

Additionally, a very quick search from my side returned a couple datasets on the topic:

·         https://data.mendeley.com/datasets/335rmgrfw2/3

·         https://deepblue.lib.umich.edu/data/concern/data_sets/mw22v557c

22)     Line 102, “By capturing full biomechanical models alongside force data on both the seat and floor,” it is unclear to me what biomechanical models implies in this context, I assume that the authors used a predefined model from the XSENS platform to estimate positional data from the IMU’s data. I would also change “force data” to “pressure data” here and I would include in the methods any calibration procedures that you adopted to calibrate the pressure mats/boards.

33)     Line 183, “a bespoke balance board” please provide specs, number and type of sensors, calibration procedures, validation, etc for this bespoke balance board.

44)     Table 1, a limitation of the study arises from the recruitment of participants that are not aged matched. This is a very important issue in the topic since people tend to perform the STS movement in different ways as they age or when they suffer from motor diseases. Since the authors present a dataset here, this should be discussed in detail. Please see articles below:

·         https://doi.org/10.1123/jab.2016-0279

·         https://doi.org/10.1080/23335432.2020.1716847

·         10.1016/j.archger.2014.12.007

·         https://doi.org/10.1016/j.jbiomech.2021.110411

55)     Line 162, “Each sensor outputs an integer value from 0 to 1024, at 4±1 Hz.” Please detail how the output from an integer value raging from 0 to 1024 was translated to force data (calibration procedures, post-processing etc). Additionally, 4Hz seems to be very low for a sampling rate for the purposes of this study (or the development of a dataset), particularly when the authors appear to have up-sampled the data to 100hz to match the sampling rate of the IMUs (line 191 – although it is not clear here if the authors only interpolated data from their balance board).

66)     Figure 1, is appears that the authors constrained the initiation of the task by placing tape on the shoes and floor. Please describe in a line or two in the methods.

77)     Line 176 “but to avoid contacting external surfaces to provide extra leverage during the STS trials.” This statement contradicts the previous statement that the authors allowed participants to perform the movement by using their arms in a preferred manner: “Instead, in this study we allowed free motion of the arms.” This should be further discussed and it should be added in the limitations of the study given that elderly and people with pathologies of the lower limbs tend to very often push the chair or their knees to stand up (please also refer to citations on my 4th comment). In the context of a robotic assistive device, this should be made clear and discussed in the limitations of the study, since, with this approach, it would be very difficult to predict the motion/input of the arms and provide appropriate assistance.

88)     Line 184 “to avoid extra strain on participants during sit-to-stand, and less velocity during stand-to-sit actions.” Since the majority of the subjects are young healthy people, including lower seat heights would have been recommended.

99)     Line 185 and throughout the text “5 STS”, I would advise to change this acronym to STSTS, from my understanding, the studied motion was the sit-to-stand-to-sit here.

110) Line 192, did the IMU sensors transmit data via a wire (it appears so from the figure)? Please specify.

111) Line 197 “and was based on the neck position on the assumption that…” the authors should elaborate more on why the opted to use the mid shoulder trajectory, instead of, for example, using the CoM, and within the scope of using those predictions for a robotic assistive device. I would argue that the CoM trajectory would have been more sensitive in this context and would capture the repositioning of the arms and legs prior to seat-off. Please also indicate here that you used the estimations of the shoulder’s positions from XSENS platform for this analysis.

112) Lines 200-204, my biggest concern about this study is methodological and it is about the averaging of the trajectories from 10 different trials. By averaging dissimilar trajectories (people may have used different strategies to stand up and sit down, or they may have performed the movement faster or slower) the variability of the movement is naturally lost. Perhaps the authors should include here appropriate references of studies that followed a similar post-processing process.

113) Line 226, why did you opt to test only for height-weight, and not also for age and pathological side for the stroke patients? It seems that age and health status are the main contributors of how people stand up and sit down.

114) Following my previous comment, it may have been better to normalise the predicted trajectories to body height or BMI in order to generate a scaled estimation of a trajectory that would fit all subjects?

115) Line 231 “predicted trajectory” please change to “predicted mid-shoulder trajectory.”

116) Line 382, “This study successfully used a k − NN and linear regression model to produce accurate trajectories” given the r2 values on table 3 and curve fits on figure 6, I wouldn’t agree that the predictions are accurate.

117) Line 387, I would change “full trajectories” to “mid-shoulder trajectory”

Reviewer 4 Report

Thank you for the opportunity to review this manuscript. 

Recommenations:

Discussions must be rewritten and extended. In discussions, the manner or elements that have been confirmed or refuted by the present study in relation to previous studies are usually identified.

The conclusions need to be extended by highlighting some relevant aspects identified in the research.

Round 2

Reviewer 3 Report

The authors have satisfactorily addressed most of my concerns.

Reviewer 4 Report

The authors improved the manuscript according with the recommendations.